# Visual Grounding Meets Language: CeAS and RAG for Bengali Long-Range Video Reasoning

## Abstract

Long-range video question answering (VQA) remains a challenging task, especially in low-resource languages like Bengali, due to limited linguistic tools and the need for multi-step temporal reasoning. To address these challenges, we propose a training-free framework for Bengali Long-range Video Reasoning (BLrVR). Our approach adapts the EgoSchema benchmark to Bengali through high-quality translation and contextual validation. We introduce a novel prompting strategy, CeAS (Close-ended Answer Selection), which integrates structured roles, task cues, and strict constraints to guide LLM reasoning. Additionally, we explore a Retrieval-Augmented Generation (RAG) variant that fuses relevant caption context with external evidence for enriched inference. Empirical results show that CeAS achieves state-of-the-art performance, surpassing RAG in precision, recall, and runtime efficiency, despite matching in accuracy and F1-score. We further benchmark different captioners, LLMs, retrievers, and prompting schemes, providing a comprehensive evaluation of components crucial to BLrVR success. Our findings demonstrate that structured prompting can outperform retrieval-heavy methods in both effectiveness and efficiency for low-resource multimodal reasoning. The **code** is publicly released at: Anaxy Code/Bengali Long Range Video Reasoning

## 1 Introduction

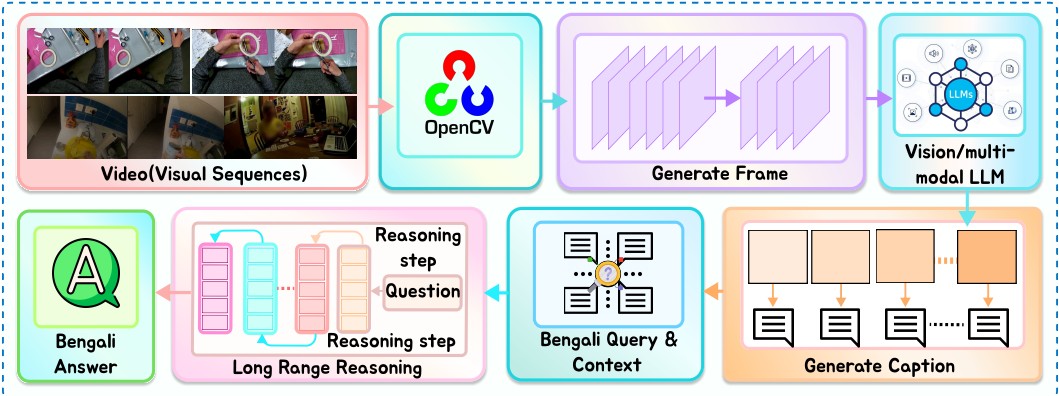

Figure 1: **End-to-end architecture for Bengali video-based question answering. The system ingests raw video sequences, extracts visual frames using OpenCV, generates captions via a vision-language model, formulates contextual Bengali queries, performs multi-step reasoning, and outputs answers in Bengali.**

Recent years have witnessed remarkable progress in short video understanding (5-15s in length) (Wang et al., 2022b; Ye et al., 2023; Fu et al., 2021; Yang et al., 2022; Wang et al., 2023e). However, extending these models to long videos (e.g., several minutes or hours in length) are not trivial due to the need for sophisticated long-range temporal reasoning capabilities. Most existing long-range

First, given a long video input, we segment it into multiple short clips and convert them into short textual descriptions using multi-modal LLM (e.g., gemini-2.0-flash-lite, gemini-1.5-flash, gemini-2.0-flash Team et al. (2023)).

video models rely on costly and complex long-range temporal modeling schemes, which include memory queues (Wu et al., 2022; Chen et al., 2020; Lee et al., 2021), long-range feature banks (Cheng & Bertasius, 2022; Zhang et al., 2021), space-time graphs ( Wang et al., 2021), state-space layers (Islam & Bertasius, 2022; Islam et al., 2023; Wang et al., 2023a) and other complex long-range modeling modules (Bertasius et al., 2021; Yang et al., 2023).

Recently, Large Language Models (LLMs) have shown impressive capability for long-range reasoning on a wide range of tasks, such as document understanding (Sun et al., 2023; Wang et al., 2023d; Gur et al., 2023) and long-horizon planning (Liu et al., 2023; Hao et al., 2023; Song et al., 2023). Motivated by these results in the natural language and decision-making domain, we explore using LLMs for Bengali long-range video question answering. The significant contributions of our proposed framework, BLrVR are as follows:

- Adapted **Bengali-translated version of the EgoSchema dataset** consisting of video, questions and answer triplets.
- We conduct an empirical study to investigate the factors behind our framework's success including (i) the selection of Reasoning Framework (ii) the selection of multi-modal LLM as captioner, (iii) the choice of an LLM, (iv) the LLM prompt design, and (v) the selection of Retriever.
- We introduce two **reasoning frameworks**, including (i) Prompting-based Reasoning Framework, and (ii) RAG-based Reasoning Framework.
- A novel structured prompt **CeAS** is proposed in research, which combines Role Specification, Task Description, Contextual Input, Structured Answer Format, and Strict Constraints.
- We introduce a multi-round summarization prompt which first instructs the LLM to summarize short-term visual captions, then answer the questions.
- Afterwards, we concatenate the temporally ordered captions by multi-modal LLM and feed them into an LLM (e.g., gemma2, gemini-pro, gemini-flash) to perform long-range reasoning for **BLrVR**.

Our framework is simple, effective, and training-free. Furthermore, it is agnostic to the exact choice of multi-modal LLM as a visual captioner and an LLM, which allows it to benefit from future improvements in visual captioning and LLM model design. We hope that our work will encourage new ideas and a simpler model design in BLrVR.

## 2 Related Work

With the emergence of LLMs, there has been an increasing research emphasis on LLM prompt design. To eliminate the need for extensive human annotations, (Kojima et al., 2022; Wang et al., 2023c) proposed zero-shot prompting methods. Subsequent research (Zhou et al., 2022; Zhang et al., 2022; Pryzant et al., 2023) has concentrated on the automatic refinement of prompts. The examination of existing works and key contributions is summarized in Figure 6.

By incorporating external, reliable data sources, Retrieval-Augmented Generation (RAG) (Gao et al., 2023; Qu et al., 2024a; Qu et al., 2024b) has recently become a viable technique for increasing the factual foundation and reasoning abilities of large language models (LLMs). For instance, RAG has been applied to tasks like report generation (Kumar & Marttinen, 2024 (Ipa et al., 2025); Tao et al., 2024) and visual question answering (VQA) (Yuan et al., 2023).

**Research Gap and Questions:** In the context of Long-Range Video Question-Answering in the Bengali language, the literature review highlights a significant research gap, particularly in video

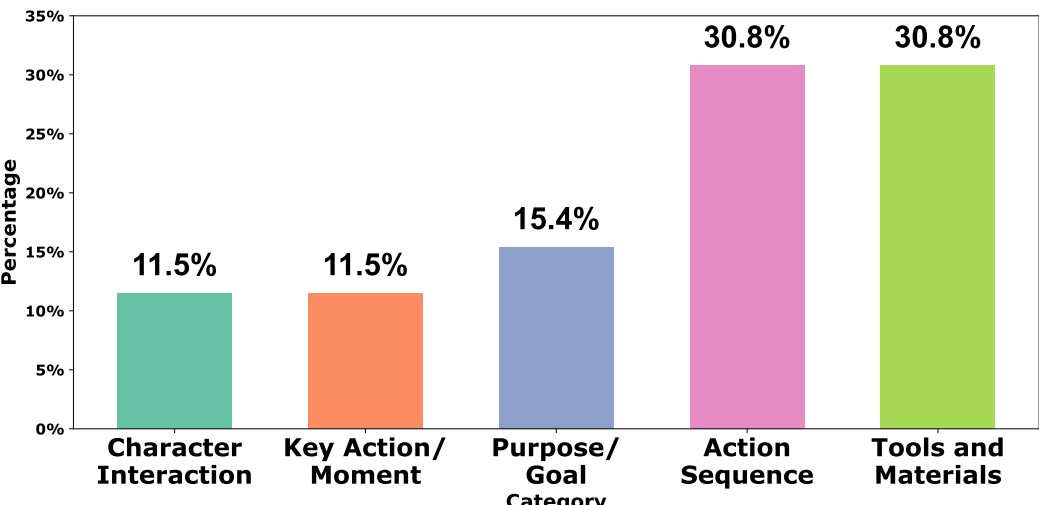

Figure 2: **Distribution of question categories in long-range video reasoning tasks**

> 💡 **Key Research Questions**
>
> - How effective are multimodal Language Models (MLMs) in video captioning on the Bengali Language?
> - Does the use of a Retrieval & Augmented Framework improve the reasoning power of BLrVR compared to using a Prompting-based Framework?
> - What are the most efficient approaches of LLMs integration for long-range reasoning in the Bengali language?

captioning quality, language biases, long-range reasoning, and the integration of LLMs for Bengali-language video understanding. This research focuses on addressing these three key research questions identified through the analysis of existing literature.

To bridge this gap, our paper introduces BLrVR, an advanced LLM-driven Long Range Video Question Answering system. This model aims to enhance Bengali video understanding by integrating optimized visual captioning and scalable LLM-based reasoning. BLrVR offers a novel and robust solution for long-range video comprehension in the Bengali language.

## 3 Proposed Methodology

This section presents the methodology used for video question-answering in the Bengali language. Our method, named BLrVR, consists of two stages: 1) short-term video clip captioning and 2) long-range text-based video reasoning using an LLM. Figure 1 presents a detailed illustration of our high-level approach. Figure 3 shows a detailed description of the internal workflow of our proposed system, highlighting its ability to process complex questions in Bengali that require grounded video understanding and multi-hop reasoning. The system receives a video and a high-level question articulated in Bengali, e.g., about the intent behind using water in a painting process and its relevance to a broader artistic technique. The key frames are extracted from the video, and each is passed through a captioning module that produces descriptive Bengali captions that contextualize the visual content (for example, a person applies water, an artist observes from a tablet, etc.). These temporally ordered captions are then passed into a structured reasoning framework. The framework performs iterative reasoning over the sequence of captions, breaking down the question into sub-steps, integrating temporally distributed evidence, and progressively constructing a logical answer. The final output is a fluent Bengali answer that not only addresses the immediate query but

also integrates cross-frame understanding. This design enables robust, explainable video question answering in a low-resource language context.

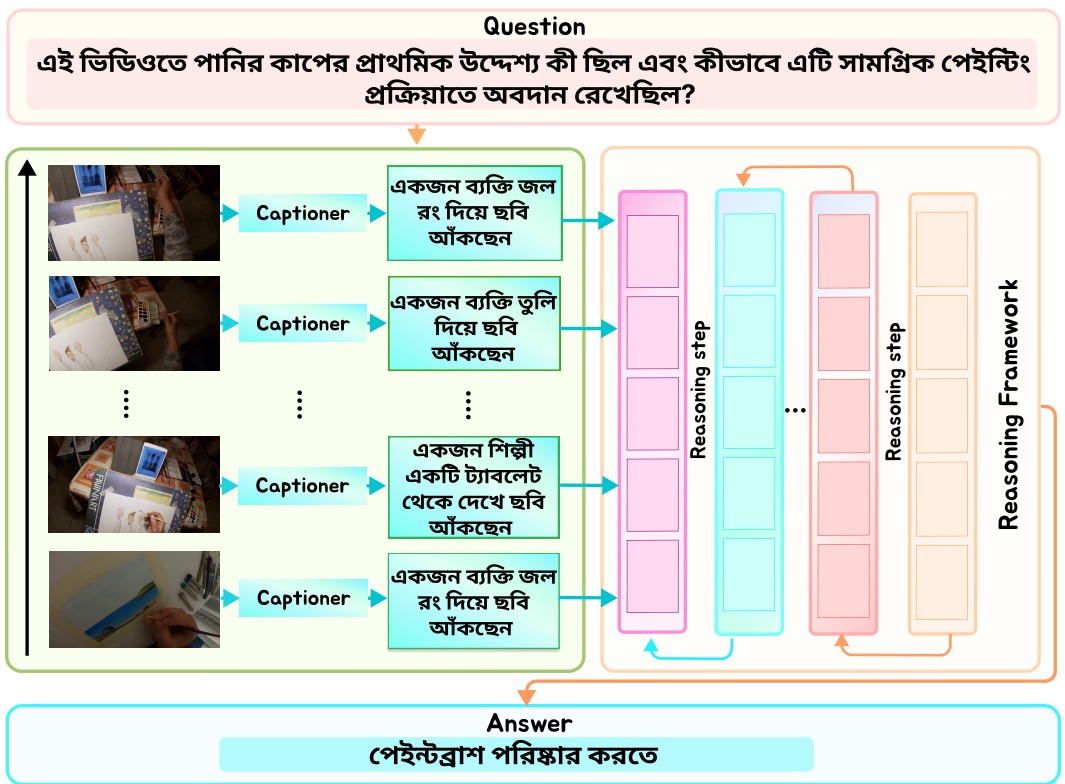

Figure 3: **Detailed workflow of Bengali video question answering system. Given a complex question in Bengali, the system extracts frames from the input video, generates frame-level Bengali captions, and passes these through a multi-step reasoning framework to derive a coherent and context-aware Bengali answer.**

### 3.1 DATASET CURATION

We have curated a custom dataset from the subset of EgoSchema Mangalam et al. (2023), a new long-range video question-answering benchmark covering a wide range of human activities. Then, we translated the Questions and the Answer Options using Google Translate[1]. Finally, we used human reasoning power to check and modify the translations accordingly to ensure that the translated text adheres to the coherence of context.

### 3.2 DATASET DESCRIPTION

Figure 2 shows the Question Category distribution. The findings indicate that there are 30.8% questions under the category Action Sequence, 30.8% under Tools and Materials, 15.4% in Purpose/Goal, 11.5% in Character Interaction, and 11.5% in Key Action/Moment.

### 3.3 SHORT-TERM VIDEO CLIP CAPTIONING

Given a long untrimmed video input $V$, we first segment it into $N_v$ non-overlapping short video clips $v = \{v_m\}_{m=1}^{N_v}$, where $v_m \in \mathbb{R}^{T_v \times H \times W \times 3}$ and $T_v, H, W$ are the number of frames, height, and width of a short video clip, respectively.

---

[1]https://translate.google.com/

Table 1: **Accuracy of different multi-modal LLM as visual captioners.**

| Multi-Modal LLM | Caption Type | Acc. (%) |
|---|---|---|
| gemini-2.0-flash Team et al. (2023) | `clip-level` | **69.2** |
| gemini-1.5-flash Team et al. (2023) | `clip-level` | 65.4 |
| gemini-2.0-flash-lite Team et al. (2023) | `clip-level` | 61.5 |

Table 2: **Accuracy of different LLMs in prompting-based reasoning.**

| LLM | Model Size | Bengali Pre-Trained | Acc. (%) |
|---|---|---|---|
| gemini-2.0-flash Team et al. (2023) | N/A | ✓ | **69.2** |
| gemini-1.5-pro Team et al. (2024a) | N/A | ✓ | 61.5 |
| gemma2-9b-it Team et al. (2024b) | 9B | ✗ | 34.6 |

Afterward, we feed each video clip $v_m$ into a pretrained short-term visual captioner $\phi$, which produces textual captions $c_m = \phi(v_m)$, where $c_m = (w_1, \ldots, w_{L_m})$ and $w_i$ represents the $i$-th word in caption $c_m$ of length $L_m$.

Note that our model is not restricted to any specific visual captioning model. Our experimental section demonstrates that we can incorporate various multi-modal LLMs as video captioners, including (gemini-2.0-flash Team et al. (2023), gemini-1.5-flash Team et al. (2023), and gemini-2.0-flash-lite Team et al. (2023)). Next, we describe how our extracted short-term captions are processed by an LLM.

Table 3: **Different variants of multi-round summarization prompt.**

| Prompt Type | CeAS | $(C) \rightarrow S$ | $(C, Q) \rightarrow S$ | $(C, Q, A) \rightarrow S$ |
|---|---|---|---|---|
| **Acc. (%)** | **69.2** | 61.5 | 61.5 | 61.5 |

### 3.4 Long-range Reasoning

We want to leverage foundational LLMs for holistic long-range video understanding. Formally, given short-term visual captions $\{c_m\}_{m=1}^{N_v}$ for all $N_v$ short video clips, we first concatenate the clip captions into the full video captions $C = [c_1, \ldots, c_{N_v}]$ in the same order as the captions appear in the original video. These concatenated captions $C$ are then processed for long-range reasoning.

## 4 Result & Discussion

We have studied the effectiveness of different components behind the success of our BLrVR framework, including (i) Visual Captioner, (ii) Answer Generator, (iii) Prompt design, and (iv) Retrieval and Augmented Generation. The experiments are conducted on the Bengali translated EgoSchema Subset with multi-choice questions. We discuss our empirical findings below.

### 4.1 Experimental Setup

The experimental setup used in this study is discussed here. Context, Questions, and Answers triplets are manipulated, pre-processed, and structured using the Pandas library, version 1.4.0, which is efficient for our reasoning task. We utilized raw Python code for natural language processing tasks to make our model evaluation tasks more efficient. We conducted our experiments on Google Colab with 15GB of memory, employing the LangChain[2] framework. This experimental setup was supported by 16 gigabytes of Random Access Memory (RAM) and 100 gigabytes of disk space, ensuring ample resources for the seamless execution of our research activities.

---

[2]https://www.langchain.com/

Table 4: **Comparison (%) of Prompting Techniques.** Performance metrics are highlighted in shades of orange, where darker cells indicate **higher values**.

| Prompting Technique | Accuracy | Precision | Recall | F1-score |
|---|---|---|---|---|
| Zero-shot CeAS (**Ours**) | **69.2** | 72.0 | 79.0 | 66.0 |
| Zero-shot Chain-of-Thought Wei et al. (2022) | 69.2 | 69.0 | 73.0 | 63.0 |
| Plan-and-Solve Wang et al. (2023c) | 61.5 | 60.0 | 69.0 | 58.0 |

---

**Algorithm 1** Prompting based Reasoning

1: **Input:** Question $Q$, Context $C$
2: **Output:** Answer $A$
3: Construct Prompt $P \leftarrow \text{Prompt}(Q, C)$
4: Initialize Language Model $LM$
5: Generate Response $R \leftarrow LM(P)$
6: Extract Answer $A \leftarrow \text{ExtractAnswer}(R)$
7: **Return** $A$

---

## 4.2 VISUAL CAPTIONING

In Table 1, we study the effectiveness of various multi-modal LLMs as clip-level video captioners, including gemini-2.0-flash Team et al. (2023), gemini-1.5-flash Team et al. (2023), and gemini-2.0-flash-lite Team et al. (2023). All baselines in Table 1 use similar experimental settings, including the same LLM model, i.e., gemini-2.0-flash for answer generation. The results are reported as BLrVR accuracy on the Bengali translated EgoSchema Subset. The results in Table 1, suggest that gemini-2.0-flash provides the best results, outperforming gemini-1.5-flash, and gemini-2.0-flash-lite.

## 4.3 ANSWER GENERATION

In this section, we have analyzed the performance of our framework based on two reasoning methods.

### 4.3.1 PROMPTING BASED REASONING

In Table 2, we analyze the performance of our framework using different LLMs while fixing the multi-modal LLM as visual captioner to be gemini-2.0-flash. Our results indicate that gemini-2.0-flash achieves the best performance (69.2%), followed by gemini-1.5-pro (61.5%). Thus, stronger LLMs (gemini-2.0-flash) are better at long-range modeling, as indicated by a significant margin in BLrVR accuracy between gemini-2.0-flash and all other LLMs (> 6.7%). We also note that the performance of gemma2-9b-it drastically degrades since it is only pre-trained on the English language. Table 2 infers that gemini-2.0-flash achieves the best accuracy, suggesting that stronger and Bengali Pre-Trained LLMs perform better in BLrVR.

**Prompt design:** In this section, we analyze several variants of our summarization-based prompt experiment with other commonly used prompt designs, including Zero-shot Chain-of-Thought

---

**Algorithm 2** Retrieval and Augmented-based Reasoning

1: **Input:** Question $Q$, Knowledge Corpus $\mathcal{D}$
2: **Output:** Answer $A$
3: Initialize Retrieval Function $f_R(Q, \mathcal{D})$
4: Retrieve Relevant Passages $R \leftarrow f_R(Q, \mathcal{D})$ from $\mathcal{D}$
5: Construct Context $C \leftarrow Q \cup R$
6: Initialize Chain Model $ChainModel$
7: Generate Intermediate Reasoning Steps $I \leftarrow ChainModel(C)$
8: Extract Final Answer $A$ from $I$
9: **Return** $A$

---

Table 5: **Performance (%) of Retrievers in Retrieval and Augmented-based Reasoning Method.**

| Retrievers | Acc. | Precision | Recall | F1-score |
|---|---|---|---|---|
| Google Embedding | **69.2** | 70.0 | 74.7 | 66.4 |
| all-mpnet-base-v2 | 65.4 | 68.4 | 70.7 | 60.3 |
| all-MiniLM-L6-v2 | 66.2 | 60.0 | 69.6 | 59.8 |
| paraphrase-multilingual-MiniLM-L12-v2 | 61.5 | 67.4 | 71.1 | 60.6 |

Table 6: **Performance (%) of Generators in Retrieval and Augmented-based Reasoning Method.**

| LLM Model | Acc. | Precision | Recall | F1-score |
|---|---|---|---|---|
| gemini-2.0-flash Team et al. (2024a) | **69.2** | 70.0 | 74.7 | 66.4 |
| gemini-1.5-flash Team et al. (2024a) | 65.4 | 61.7 | 54.7 | 55.0 |
| gemma2-9b-it Team et al. (2023) | 46.2 | 39.9 | 44.6 | 37.1 |
| gemini-1.5-pro Team et al. (2024a) | 38.5 | 52.8 | 31.3 | 32.4 |

(Zero-shot CoT) Wei et al. (2022), and Plan-and-Solve Wang et al. (2023c) (described in A). Below, we present a detailed analysis of these results.

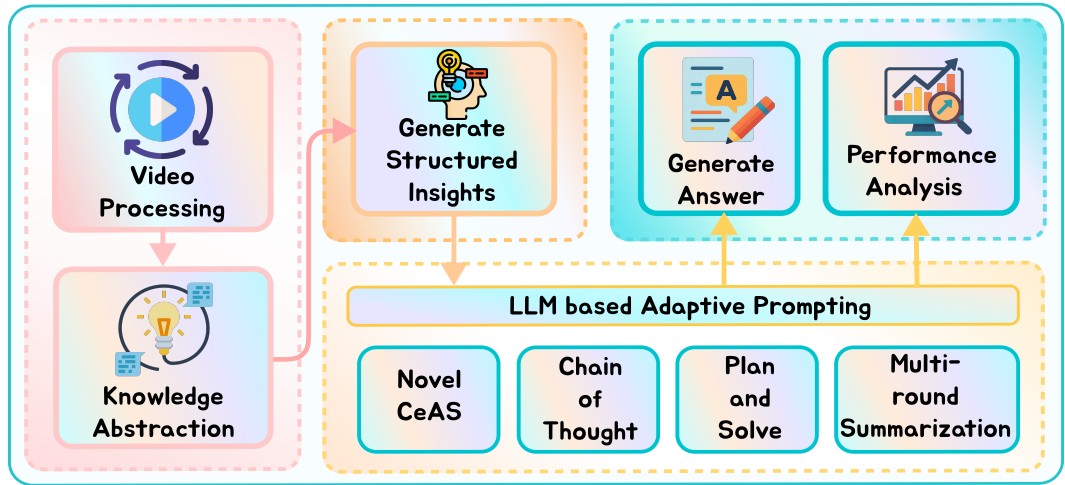

Figure 4: **Prompting based reasoning framework. This module leverages video-derived knowledge abstractions and generates structured insights, which are processed through LLM-based adaptive prompting strategies, including Chain-of-Thought, CeAS, and Plan-and-Solve to produce coherent answers and support performance analysis.**

**Multi-round Summarization Prompt:** In Table 3, we explore how these three different summarization prompts affect performance. Our results show that the CeAS (non-summarized) approach achieves the best accuracy (**69.2%**), suggesting that summarization may cause **information loss**. Specifically, we note that summarizing only captions (C) leads to 61.54% accuracy, which is a drop from the standard approach in performance (**-7.66%**). Furthermore, we observe that summarizing both captions (C) and the question (Q) does not improve accuracy beyond summarizing only captions (C). This indicates that summarization-based prompting can sometimes harm performance in BLrVR tasks.

**Comparison with Commonly Used Prompt:** Next, in Table 4, we compare our CeAS prompt with other commonly used prompts such as Zero-shot Chain-of-Thought Wei et al. (2022), and Plan-and-Solve Wang et al. (2023c). Among these commonly used prompts, the Zero-shot Chain-of-Thought prompting technique (69.2%) performs equally well as CeAS(69.2%), suggesting that reasoning-based prompting (as proposed by Wei et al., 2022) is competitive. Plan-and-Solve (61.5%)

has the lowest accuracy, indicating that this approach may be less effective for this particular evaluation compared to the other two methods. However, the Chain-of-Thought approach falls slightly behind **CeAS** in precision and F1-score, indicating a trade-off between correctness and consistency in prediction.

### 4.3.2 Retrieval and Augmentation-based Reasoning

**Retriever:** Table 5 shows retrieval performance using different retrievers (embedding models) under a Retrieval and Augmented-based reasoning Method. Our results show that **GoogleEmbedding** is the strongest retriever overall, suggesting that using more powerful embeddings significantly boosts retrieval-augmented reasoning quality. It achieves the highest Accuracy (69.2%), highest Precision (70.0%), highest Recall (74.7%), and highest F1-score (66.4%). Specifically, we observe that all-MiniLM-L6-v2 has decent Accuracy (66.2%), but very low Precision (only 60.0%) — indicating a lot of false positives, whereas all-mpnet-base-v2 is more balanced in Precision (68.4%) and Recall (70.7%) but has lower Accuracy (65.4%). Furthermore, we observe that paraphrase-multilingual-MiniLM-L12-v2 is the weakest in Accuracy (61.5%), although Recall (71.1%) is relatively high, Precision (67.4%) is not enough to compensate.

**Generator:** Table 6 evaluates different LLM Generators in a Retrieval and Augmented based method. We can see that **gemini-2.0-flash** is the strongest generator overall, achieving Accuracy (69.2%), Precision (70.0%), Recall (74.7%), and F1-score (66.4%). gemini-1.5-flash is second-best but with noticeable drop-off — small drop in Accuracy (around 4%) and sharp drops in Precision (61.7%) and Recall (54.7%) compared to 2.0-flash. It suggests that model version upgrades strongly matter; even seemingly small version changes lead to substantial real-world QA improvements.

### 4.4 Ablation Study

We compare our proposed Prompting-based approach with the state-of-the-art Retrieval and Augmentation-based approach to assess their effectiveness in BLrVR tasks. Specifically, we can compare the two best-performing prompting strategies, Chain-of-Though and our custom CeAS method, against the best-performing Gemini-2.0-Flash-based RAG method from Table 4, Table 6, and Table 9. While all approaches yield the same accuracy (69.2%), our proposed **CeAS** prompting technique shows the most balanced performance with precision (**72.0%**), recall (**79.0%**), and F1-score (**66.0%**). Although RAG-based Gemini-2.0-Flash achieves a slightly higher F1-score (66.4%), it comes with lower precision (70.0%) and recall (74.7%) compared to CeAS. Furthermore, Chain-of-Thought lags further behind with an F1-score of 63.0%. In terms of runtime, prompting-based methods (CeAS: 0.046s, Chain-of-Thought: 0.043s) demonstrate superior efficiency compared to the RAG-based approach (0.051s). These results highlight the advantage of language-adaptive prompting in low-resource languages like Bengali, offering both higher reasoning quality and lower inference cost without relying on retrieval mechanisms.

## 5 Conclusion

We introduced a lightweight yet effective framework for BLrVR, addressing the notable gap in low-resource language research. By adapting the EgoSchema dataset to Bengali and designing a structured prompting framework, we demonstrate the value of leveraging instruction-tuned LLMs for extended temporal reasoning. Our novel CeAS prompt design outperformed both multi-round summarization and widely adopted prompting techniques, achieving state-of-the-art accuracy while offering superior precision, recall, and runtime efficiency. In parallel, we explored the RAG framework, which, despite achieving comparable accuracy, incurred higher computational costs and slightly lower precision-recall balance.

Through extensive benchmarking, we showed that Bengali pre-trained LLMs, when combined with strong captioners and structured prompts, enable robust question-answering over long-form videos. Our findings highlight that language-aware prompt engineering can match and even surpass retrieval-based methods in low-resource settings. This work opens promising directions for scalable, training-free BLrVR systems and sets the groundwork for further hybridization of prompting and retrieval strategies. Future research will explore adaptive multi-step reasoning and dynamic prompt construction across multilingual long-range video understanding benchmarks.

## 6 LIMITATIONS

Our proposed framework used different open-source multi-modal LLM for the short-term visual captioning task on our custom-curated EgoSchema. Paid multi-modal LLM that is supposed to work better remains to be explored in the future. While we are open to different types of limitations, just mentioning that a set of results has been shown for English only probably does not reflect in the Bengali language. Mentioning that the existing method works mostly based on instruction in limited morphology languages, like English, but not in morphology-rich languages, like Bengali. In addition, limitations such as low scalability to long text, the requirement of large GPU resources, or other things that inspire crucial further investigation are welcome.

## ETHICS STATEMENT

We use only publicly available models and data for this research. The EgoSchema subset was manually curated and contains no personal or sensitive information. Our work aims to support low-resource language processing, and we will take caution against direct deployment without further validation, especially in critical domains.

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

Our appendix consists of Prompt-based reasoning (Section A), Retrieval and Augmented-based Reasoning (Section B), Video Processing (Section C), Additional Analysis (Section D), Additional Implementation Details (Section E), and Qualitative Analysis (Section F).

## A    Prompting based Reasoning

The concatenated video captions $C$ are fed into an LLM for long-range video reasoning. Specifically, given the concatenated video captions $C$, the question $Q$, and the answer candidates $A$, we prompt the LLM to select the correct answer using the following prompt template:

> *"Please provide a single-letter answer (0, 1, 2, 3, 4) to the following multiple-choice question {Q}. You are given language descriptions of a video. Here are the descriptions: {C}. Here are the answer options {A}."*

However, many modern LLMs may struggle when provided with long (>1K words), noisy, and potentially redundant/irrelevant caption sequences. To address these issues, we also investigate a special kind of LLM prompts that ask an LLM first to summarize the noisy short-term visual captions (first round of prompting) and then answer a given question about the video (second round of prompting).

Specifically, we formulate such a multi-round prompt as follows: given the video captions $C$, the question $Q$, and the answer options $A$, instead of directly feeding the $\{C, Q, A\}$ triplet into the LLM for BLrVR, we first ask the LLM to provide a summary of the captions in the first round, which we denote as $S$, using the following prompt template:

*"You are given language descriptions of a video in Bengali Language: {C}.*
*Please give me a {N_w} word summary."*

$N_w$ denotes the desired number of words in the summary $S$. Afterward, during the second round of prompting, instead of using the captions $C$, we use the summary $S$ as input for the LLM to select one of the answer candidates. Conceptually, such a prompting scheme is beneficial, as the LLM-generated summary $S$ filters out irrelevant/noisy information from the initial set of captions $C$, making LLM inputs for the subsequent QA process more succinct and cleaner.

In our study, we employ (1) Close closed-ended Answer Selection (CeAS) prompt,(2) several variants of our summarization-based prompt, and (3) other commonly used prompt designs, including Zero-shot Chain-of-Thought (Zero-shot CoT) Wei et al. (2022), and Plan-and-Solve Wang et al. (2023c). Figure 4 depicts the detailed architecture of the Prompting-based Reasoning Method.

**Close ended Answer Selection Prompt** Given a concatenated set of captions C, an input question Q, and a set of candidate answers A, prompt is designed to directly select the most appropriate answer $Y \in \{0, 1, 2, 3, 4\}$ without generating intermediate reasoning, explanation, or rationale.

**Multi-round Summarization Prompt** Given a concatenated set of captions C, an input question Q, and a set of candidate answers A, we can use several input combinations to obtain the summary S. Thus, here, we investigate three distinct variants of obtaining summaries S:

- **(C) → S**: the LLM uses caption-only inputs C to obtain summaries S in the first round of prompting.
- **(C, Q) → S**: the LLM uses captions C and a question Q as input for generating the summaries S. Having additional question inputs is beneficial as it allows the LLM to generate a summary S, specifically tailored for answering an input question Q.
- **(C, Q, A) → S**: the LLM takes captions C, a question Q, and the answer options A as its inputs to produce summaries S. Having additional answer candidate inputs may enable the LLM to generate a summary S most tailored to particular question-answer pairs.

**Commonly used Prompt:** We explore several advanced prompting techniques to integrate intermediate reasoning steps before deriving the final answer:

- Chain-of-Thought: Given a concatenated set of captions C, an input question Q, and a set of candidate answers A. The LLM is prompted with explicit reasoning cues such as:

  *"Let's think step by step"*

- Plan-and-Solve: This prompting technique introduces a two-stage approach: (i) Planning, where the LLM is first guided to generate a structured outline of the reasoning process, and (ii) Solving, where the model follows the generated plan to derive the final answer.

# B   RETRIEVAL AND AUGMENTED-BASED REASONING

Instead of relying on entire context $C$, this method integrates retrieval-augmented generation (RAG) to supplement most relevant context by incorporating external knowledge $R$. Specifically, we define an external retrieval function $f_R(Q)$ that retrieves relevant passages $R = \{r_1, ..., r_M\}$ from a retrieval corpus $\mathcal{D}$ using dense passage retrieval (DPR) conditioned on the question $Q$:

$$R = f_R(Q) = \text{Top-K}(\text{RetrievalModel}(Q, \mathcal{D})) \tag{1}$$

Afterwards, the updated input $\tilde{C}$, along with the question $Q$ and answer candidates $A = \{a_1, a_2, ..., a_k\}$, are dynamically integrated with an LLM to select the most probable answer:

$$\hat{a} = \arg\max_{a_i \in A} LLM(R, Q, A) \tag{2}$$

Unlike direct caption reasoning, this approach allows the LLM to utilize external and most relevant evidence, improving factual accuracy and reducing reliance on the noisy, and excessive amount of captions. Retrieval and Augmented-based Reasoning Method is depicted in Figure 5.

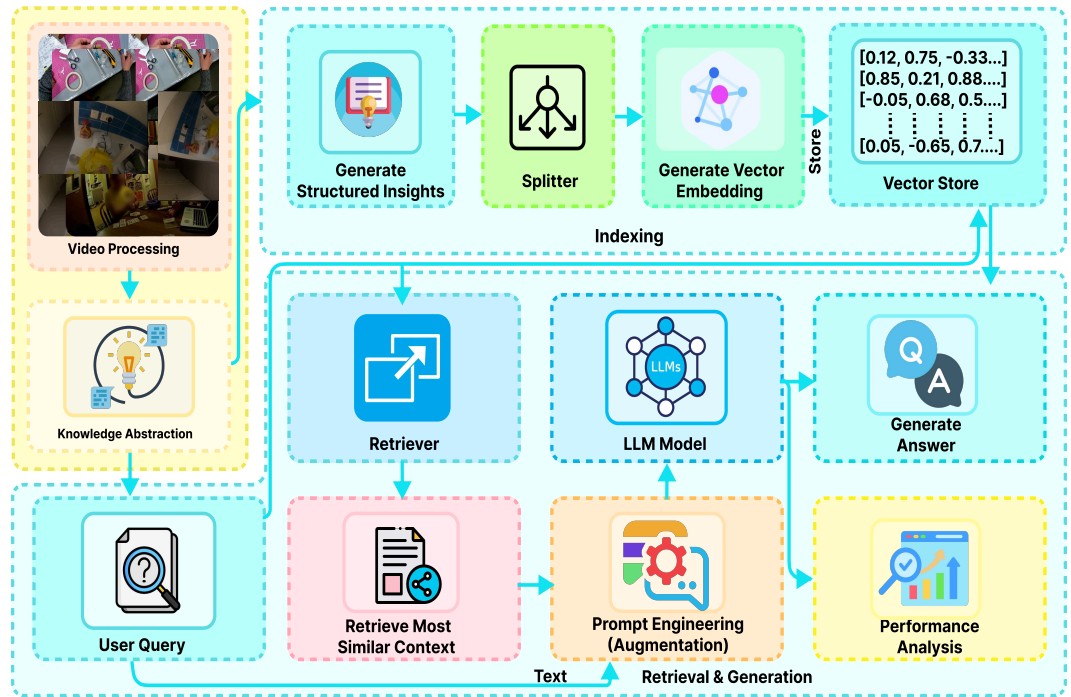

Figure 5: **Retrieval and Augmentation based reasoning framework. The system processes video-derived insights, generates vector embeddings, and stores them in a vector database. Given a user query, it retrieves the most relevant context, augments it via prompt engineering, and uses an LLM to generate Bengali answers with integrated performance analysis.**

## C   VIDEO PROCESSING

Each video is 3 minutes length, therefore, we segment each video into 180 numbered 1s clips with a strideof 1s using OpenCV[3], resulting in a list of consecutive clips that cover the entire video.

## D   EXISTING WORK DETAILS

Recent advances in video understanding have spurred research into modeling long-range temporal dependencies, integrating large language models (LLMs), and improving reasoning for video question answering (VidQA). Existing literature can be broadly categorized into four areas: long-range video modeling, LLM-based video understanding, VidQA benchmarks, and LLM prompt engineering.

### D.1   LONG-RANGE VIDEO UNDERSTANDING

Traditional video models often struggle with capturing extended temporal context. Sun et al. (2022) proposed LF-VILA, which utilizes Hierarchical Temporal Window Attention (HTWA) to model temporal features across multiple scales. Memory-augmented methods such as MeMViT (Wu et al., 2022) and MovieChat (Song et al., 2024) enhance long-term retention by leveraging memory units for sequential context aggregation. Structured state-space models, including S4ND (Nguyen et al., 2022), ViS4mer (Islam & Bertasius, 2022), and S5 (Wang et al., 2023a), have emerged as alternatives to transformers by efficiently modeling long-range dependencies using recurrent state dynamics.

---

[3]https://opencv.org/

## D.2 LLMs for Video Understanding

Several studies explore the integration of vision encoders with LLMs. Socratic Models (Zeng et al., 2022) and VideoChat (Li et al., 2023) align visual and textual modalities to perform multimodal reasoning. Models like Video ChatCaptioner (Chen et al., 2023) and ChatVideo (Wang et al., 2023b) further enable interactive, dialogue-based video understanding. VidIL (Wang et al., 2022c) adapts image-based LLMs for video tasks using few-shot learning. Although some recent approaches (Lin et al., 2023; Bhattacharya et al., 2023) extend LLMs to long-form video scenarios, they lack strong quantitative evaluations to validate effectiveness.

## D.3 Video Question Answering (VidQA)

Datasets like How2QA (Yang et al., 2021) support short and long-range VidQA tasks but often rely heavily on textual transcripts, limiting their assessment of true visual reasoning. EgoSchema (Mangalam et al., 2023) addresses this by requiring analysis of videos over 100 seconds and minimizing language bias, offering a more realistic testbed for visual reasoning.

## D.4 LLM Prompt Design for Video Tasks

Prompting strategies have been crucial in enhancing LLM performance on video reasoning tasks. Chain-of-Thought prompting (Wei et al., 2022) improves answer quality by encouraging step-by-step reasoning. Plan-and-Solve (Wang et al., 2023c) decomposes complex queries into sub-tasks, while Self-Consistency (Wang et al., 2022a) increases reliability by aggregating answers from multiple inference rounds.

A summary of representative methods and their key contributions across the discussed categories is presented in Figure 6.

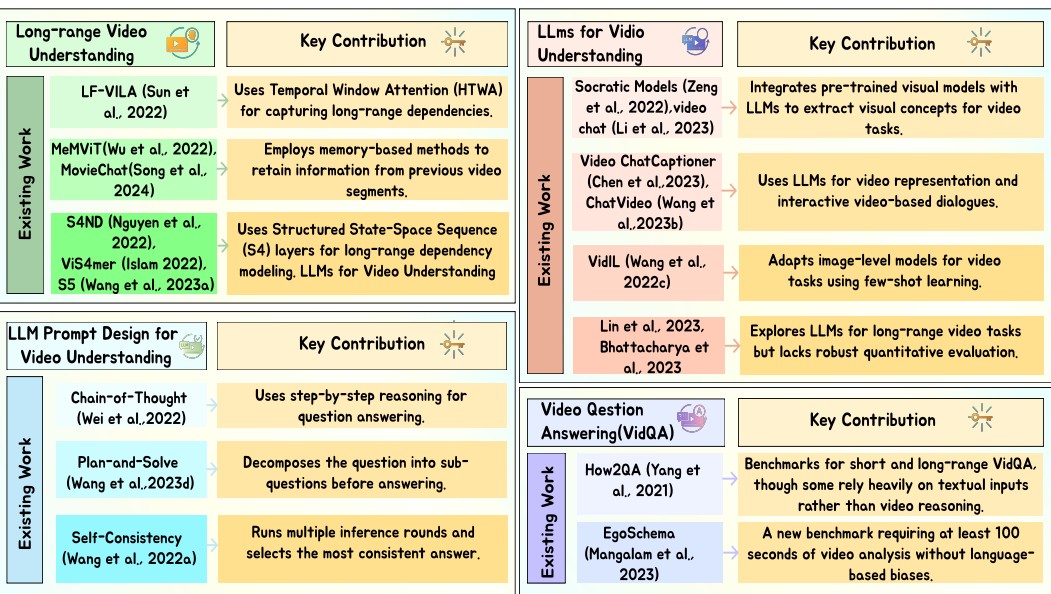

Figure 6: **Summary of existing work and key contributions across long-range video understanding, LLM-based video reasoning, video QA, and prompt design.**

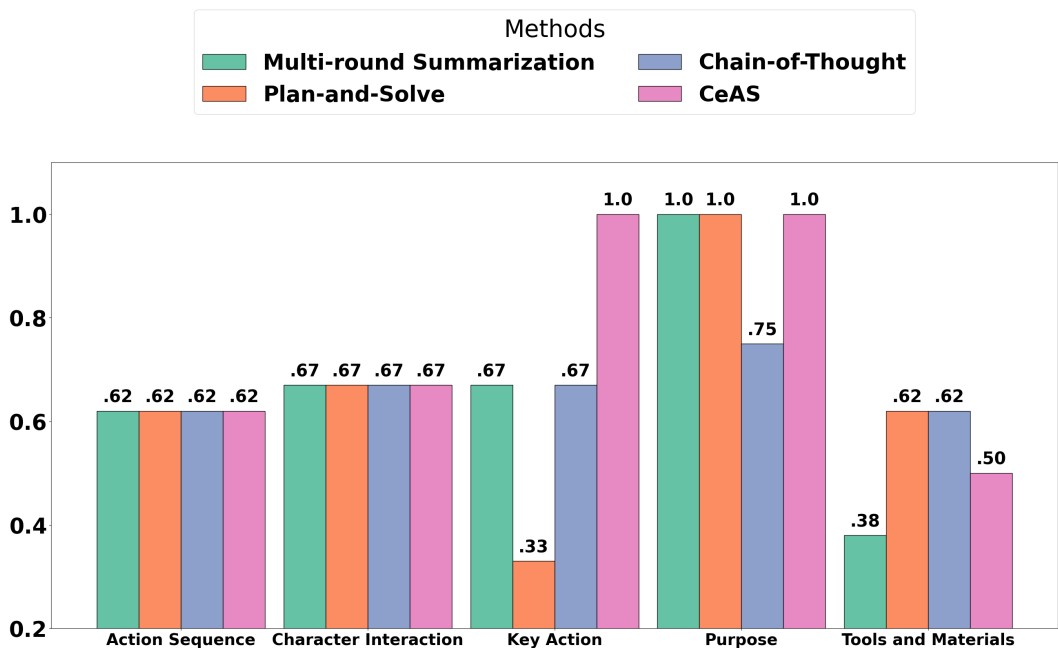

Figure 7: **Accuracy of question category-based answers in Prompting-based reasoning.**

# E    ADDITIONAL ANALYSIS

## E.1    EVALUATION METRICS

To assess the effectiveness of our BLrVR approach, we employ commonly used evaluation metrics: Accuracy, Precision, Recall, and F1-score. These metrics help quantify the quality and correctness of the generated answers.

$$\text{Accuracy} = \frac{\text{Correct Answers}}{\text{Total Answers}} \tag{3}$$

$$\text{Precision} = \frac{\text{Relevant and Correct Answers}}{\text{Generated Answers}} \tag{4}$$

$$\text{Recall} = \frac{\text{Correct Answers}}{\text{Expected Correct Answers}} \tag{5}$$

$$\text{F1-score} = 2 \times \frac{\text{Precision} \times \text{Recall}}{\text{Precision} + \text{Recall}} \tag{6}$$

These metrics provide a concise yet effective evaluation framework for measuring answer quality of our BLrVR approach.

## E.2    ACCURACY ON DIFFERENT QUESTION TYPES

In Figure 7, we break down the performance of different prompting methods across various question categories. Our results indicate that all the methods perform consistently in the Character Interaction and Action Sequence categories (around 62%–67% accuracy). One possible explanation is that these categories may require a relatively straightforward understanding of sequential activities and simple interactions, which all methods can handle equally well.

We also observe that the Plan-and-Solve method struggles significantly in the Key Action/Moment Detection category (33% accuracy), whereas the CeAS method achieves perfect performance (100%)

in this category. We conjecture that Plan-and-Solve may fail due to limitations in isolating key moments across long temporal contexts, while CeAS is more effective at pinpointing critical events.

In the Purpose/Goal Identification category, CeAS, Chain-of-Thought, and Multi-round Summarization all reach 100% accuracy, highlighting that these methods can effectively infer human intentions when sufficient context is available.

Lastly, while performance in the Tools and Materials category is generally lower (ranging from 38% to 62%), it remains encouraging that Chain-of-Thought and Multi-round Summarization still maintain relatively robust accuracy compared to the others. These results demonstrate that while some categories remain challenging, strong prompting strategies like CeAS can achieve near-optimal understanding across complex video question categories.

# F    Additional Implementation Details

## F.1    Visual Captioning

For the final experiments, we use gemini-2.0-flash Team et al. (2023) as our multi-modal LLM captioner. We design a standard Gemini prompt to generate captions with roughly 10 words for each frame to avoid the content-length exceeded error for next-level LLMs' answer generation based on reasoning capability.

To get rid of the extreme repetitiveness of pure greedy decoding, we have used a combination of top-k sampling and nucleus sampling with k = 5 and p = 0.95. Then we take the candidate with the largest confidence score as the final caption of the video clip. Specifically, we use this prompt:

> *"<image>. Act as an expert in Image Captioning. Your task is to generate accurate and precise caption in the Bengali Language to describe the image properly. Caption should be in one sentence within 10 words"*

## F.2    Caption Processing

When generating a visual caption, LLM has outputted additional text alongside the caption despite specifying the instruction. Therefore, post-processing like pattern-based extraction, removal of special characters, and additional punctuation removal has been implemented. Before-after of post-processed prediction is shown in Table 7.

We can see that the model outputs often contain auxiliary framing text (e.g., "Here's a caption in Bengali…"), multilingual annotations (e.g., Romanized Bengali - "Kāṭhera kājera jāygā" or English translations - "Woodworking area"), and formatting artifacts (e.g., parentheses, markdown, or prompt templates). Post-Processing step significantly improves caption readability and usability, especially in low-resource settings where raw outputs often contain hallucinated or templated noise.

## F.3    Large Language Models(LLMs)

For most experiments on curated EgoSchema, we use gemini-2.0-flash as the LLM. Specifically, we use the gemini-2.0-flash variant, which is optimized for fast inference. When needed, we also evaluate gemini-1.5-flash to analyze performance under the same LLM settings. We set the generation temperature to 0 for all experiments to maintain deterministic behavior.

For additional comparisons, we experiment with gemma2-9b-it and gemini-1.5-pro models. Across all models, to prioritize factual consistency, we have taken the candidate with the largest confidence score as the final output following a combination of top-k sampling and nucleus sampling with k = 5 and p = 0.95. Unless otherwise specified, gemini-2.0-flash remains our primary choice due to its superior accuracy and efficiency trade-off.

### F.4 Answer Generation in Prompting-based Reasoning

For the final experiments on answer generation, we use gemini-2.0-flash Team et al. (2023) as the Prompting-based Reasoning LLM. We use 0 as the temperature for all experiments. We use gemini-1.5-pro Team et al. (2024a) as Gemini Pro variants. As the Gemma model, gemma2-9b-it Team et al. (2024b) is used.

**Output Processing:** When answering multiple choice questions, LLMs usually output complete sentences instead of a single-digit answer, i.e., 0, 1, 2, 3, or 4. One way to obtain the single-character response is to perform post-processing on the output, which usually requires substantial engineering efforts like explicitly prompting as in Figure 8.

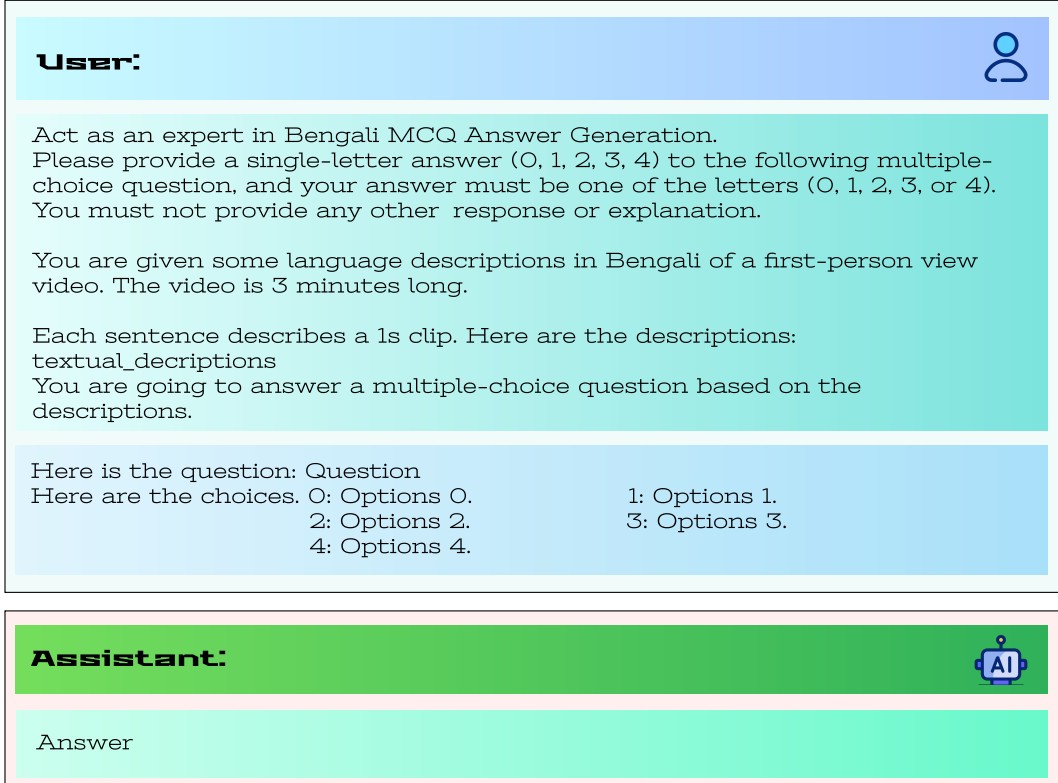

Figure 8: **Prompt in prompt based reasoning**

The prompt instructs the model to behave as an expert in Bengali MCQ answering. It specifies that the model must output a single-letter answer (0, 1, 2, 3, or 4) corresponding to one of the given options without any additional explanation. The input to the model consists of:

- Bengali language descriptions from a 3-minute first-person view (FPV) video, where each sentence describes a 1-second video clip.
- A multiple-choice question along with five answer options (labeled 0 to 4).

The Assistant's expected behavior is strictly constrained to output only the selected answer number, ensuring focused, minimalistic responses optimized for automatic evaluation.

### F.5 Retrieval and Augmentation-based Reasoning

For the final experiments on answer generation, we use gemini-2.0-flash Team et al. (2023) as the Retrieval and Augmented based Reasoning LLM in Figure 9. We use 0 as the temperature for all experiments.

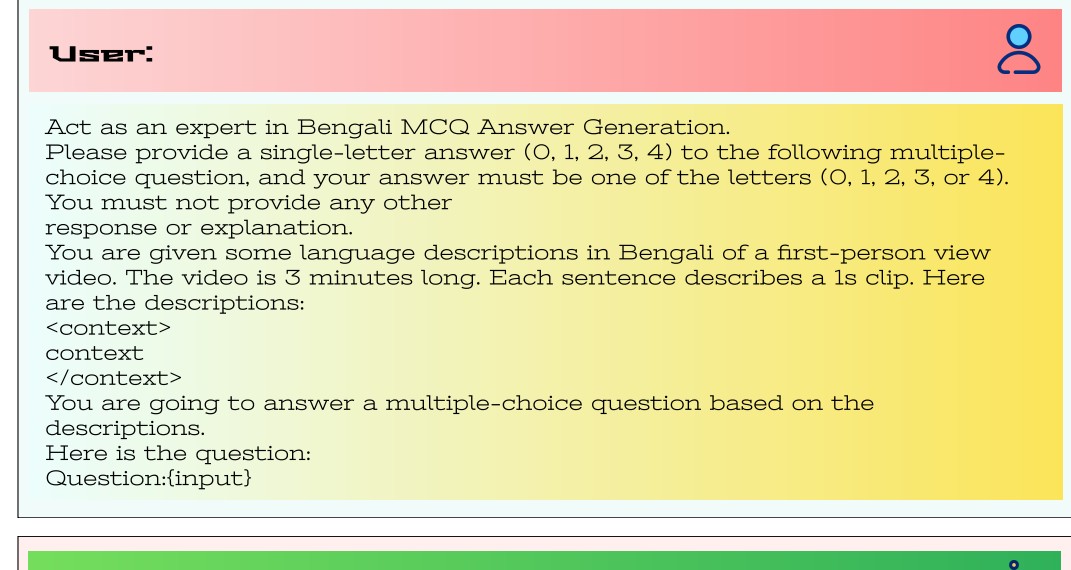

Figure 9: **Prompt in retrieval and augmented based reasoning**

We use gemini-1.5-pro Team et al. (2024a) as Gemini Pro variants. As the Gemma model, gemma2-9b-it Team et al. (2024b) is used. gemini-1.5-flash Team et al. (2023) is also used for experimenting.

**Output Processing** As LLMs prefer generating complete output sentences, we design explicit prompts as in Figure 9 for Retrieval and Augment based Reasoning to force LLMs to generate a single character as the response. The prompt instructs the model to act as an expert in Bengali multiple-choice question answering, with strict constraints:

- ✔ The model must provide a single-letter answer (0, 1, 2, 3, or 4) corresponding to the given choices.
- ✔ No explanations or additional text are permitted beyond the selected answer.

The input consists of:

- ✔ Contextual descriptions in Bengali, where each sentence corresponds to a 1-second clip from a 3-minute first-person view (FPV) video.
- ✔ A multiple-choice question (input) based on the provided descriptions.

The <context> tag encapsulates the list of Bengali descriptions to help maintain clear boundaries between context and question.

The Assistant's expected behavior is to directly output the final answer, ensuring a highly structured and automatable evaluation pipeline.

Although the model is instructed to follow a specific output format, it does not consistently comply. To address this inconsistency and support reliable evaluation, particularly QA benchmarks, we introduce an additional post-processing pipeline. This pipeline is designed to distill the model output into a clean and structured scalar form (e.g., 4). The process involves applying pattern matching to extract the final predicted option. These post-processing steps ensure consistency, improve readability, and facilitate accurate automatic evaluation. Table 8 illustrates representative examples of generated outputs before and after additional post-processing, demonstrating the effectiveness of this approach in producing clean and interpretable results.

Table 7: **Before-after of post-processing**

| Generated Caption | Post-Processed Caption |
|---|---|
| Here's a caption in Bengali, following the specified constraints:
কক্ষটির মেঝে সংস্কারের কাজ চলছে। | কক্ষটির মেঝে সংস্কারের কাজ চলছে। |
| Here's a caption in Bengali, following the instructions:
কাঠের কাজের জায়গা।
(Kāṭhera kājera jāygā.)
(Woodworking area.) | কাঠের কাজের জায়গা। |
| Here's a caption in Bengali, following the instructions:
**অনেক ঘাস কাটার যন্ত্রপাতি ও ব্লোয়ারের ছবি।** | অনেক ঘাস কাটার যন্ত্রপাতি ও ব্লোয়ারের ছবি। |

Table 8: **Before-after of additional post-processing**

| Predicted Output | Additional Post-Processed Output |
|---|---|
| 1. **উপযুক্ত তথ্য নিষ্কাশন:**
* ভিডিওতে হাতে আঁকা ছবি, ল্যান্ডস্কেপ পেইন্টিং, সমুদ্র সৈকতের ছবি, ঘর ও সমুদ্রের ছবি ইত্যাদি বিভিন্ন বিষয়বস্তু দেখা যাচ্ছে। ...
2. **উপপ্রশ্ন তৈরি:**
* প্রশ্ন ১: ভিডিওতে কি শিল্পী বিভিন্ন সময়ে বিভিন্ন উপকরণ (যেমন: তুলি থেকে কলম) ব্যবহার করেছেন? যদি করে থাকেন, তবে এর কারণ কী হতে পারে ? ...
4. **চূড়ান্ত উত্তর:**
বর্ণনার উপর ভিত্তি করে, ৪ নম্বর পছন্দটি সবচেয়ে উপযুক্ত।
অতএব, উত্তর: 4 | 4 |
| Here's my thought process:
1. **Relevant Information Extraction:**
* The descriptions mention "রুটি তৈরি" (making bread), "মাটি চাষ" (cultivating soil), "একটি পাত্রে কিছু রান্না হচ্ছে" (something is being cooked in a pot), ...
* There are also descriptions of kitchen scenes, including the sink, counter, refrigerator, and shelves with various items. ...
4. **Answering the Multiple Choice Question:**
Based on the descriptions, the person is engaged in the overall process of cooking a meal ...
Therefore, the answer is: 4 | 4 |

# G  QUALITATIVE ANALYSIS

## G.1  CAPTIONERS

Figure 10 presents a qualitative comparison of the captions generated by gemini-2.0-flash and gemini-1.5-flash across diverse video frames. We observe that gemini-2.0-flash consistently produces more detailed, action-centric descriptions, whereas gemini-1.5-flash tends to generate broader and less specific captions. For instance, in the first frame, gemini-2.0-flash accurately captures the fine-grained action ("টেবিল স দিয়ে কাঠ কাটা"), while gemini-1.5-flash merely notes the presence of the tile without recognizing the associated action ("টেবিল স দেখানো").

Similar trends are evident in subsequent frames, where gemini-2.0-flash effectively grounds physical interactions (e.g., "সরঞ্জাম ধরে রাখা", "লাল লন মাওয়ার স্থাপন") compared to the more generic descriptions produced by gemini-1.5-flash. These results highlight the superior grounding capability of gemini-2.0-flash, which is crucial for enhancing downstream tasks that rely on fine-grained visual understanding and precise language grounding, such as complex reasoning and video question answering.

Table 9: **Runtime of prompting-based and retrieval-augmented reasoning methods**

| Reasoning Method | Prompt | | | |
| --- | --- | --- | --- | --- |
| **Prompting-based** | CeAS | Chain-of-Thought | Plan-and-Solve | Multi-round Summarization |
| Runtime | 0.046 | 0.043 | 0.039 | 0.2818 |
| **Reasoning Method** | Retrieval | | | |
| **Retrieval & Augmented** | Google Embedding | all-mpnet-base-v2 | all-MiniLM-L6-v2 | paraphrase-multi-lingual-MiniLM-L12-v2 |
| **Runtime** | 0.051 | 17.88 | 2.7 | 5.88 |

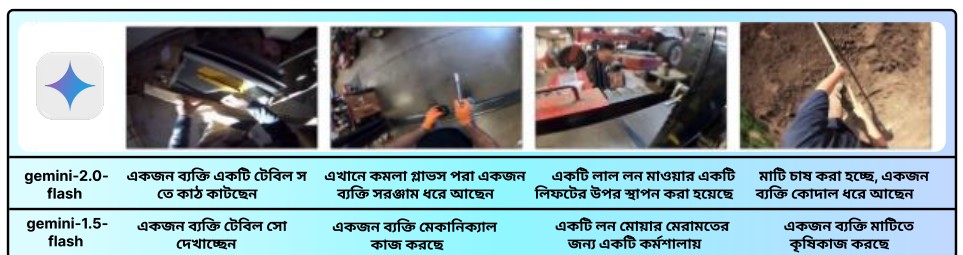

Figure 10: **Bengali video captions generated by Gemini-2.0-flash and Gemini-1.5-flash**

## G.2 LVRQR WITH PROMPTING-BASED REASONING

Figure 11 presents a successful example from our Prompting-based Reasoning Framework. Our proposed CeAS effectively demonstrates the model's ability to reason over the visual-textual context. It correctly identifies that the subject is engaging in a soil manipulation task and employs a rake as the primary tool. The rationale integrates multiple observations — such as "মাটিতে কাজ করা", "উভয় হাত দিয়ে রেক ধরে রাখা", and "মাটি কোপানো" — to conclude that the rake was used to level the ground. This example highlights the strength of our prompting approach in combining grounded visual understanding with long-form reasoning, which is particularly effective for complex compositional queries in instructional videos.

Figure 12 illustrates a failure case from our Prompting-based Reasoning Framework when answering a question involving complex causal inference regarding safety and precision. The question targets safety and precision, but the system fails to link actions to higher-order reasoning. This highlights a key limitation: while it describes visible actions well, it struggles with implicit intentions, pointing to the need for better common sense and causal reasoning.

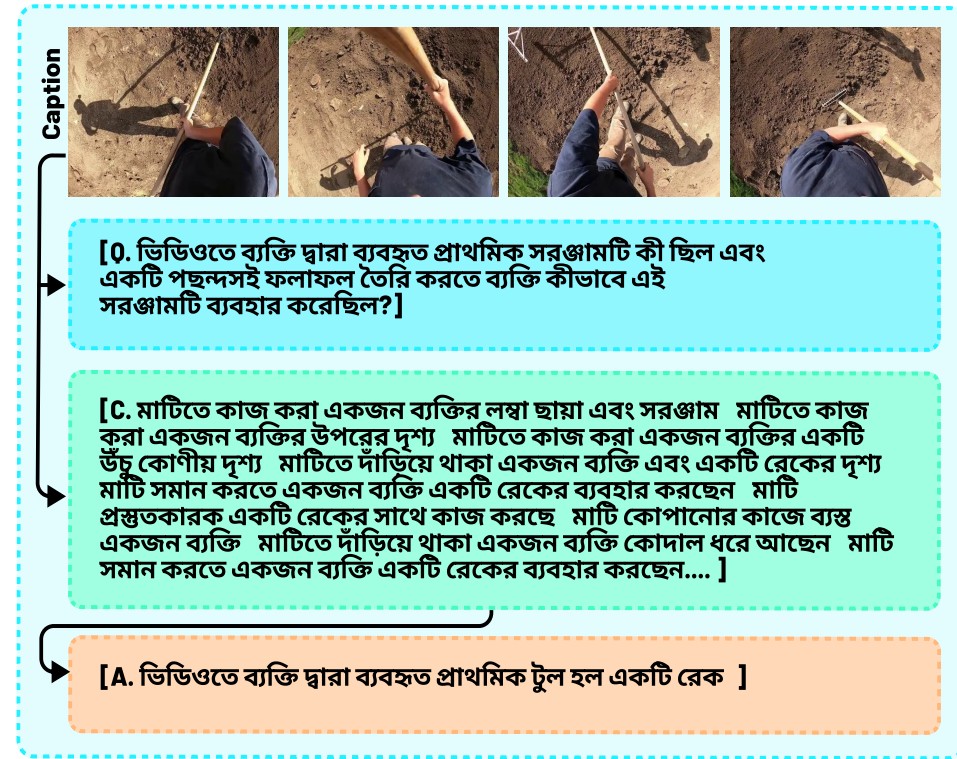

Figure 11: **Success case of our prompting-based reasoning framework**

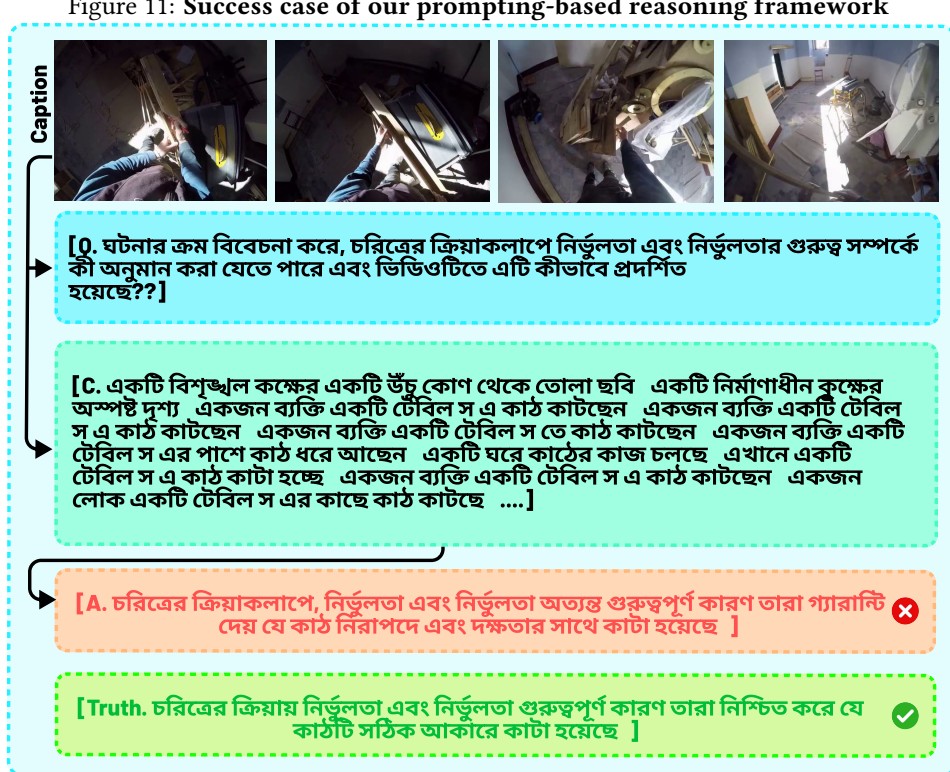

Figure 12: **Failure case of our prompting-based reasoning framework**

