# OpenReview forum: "Visual Grounding Meets Language: CeAS and RAG for Bengali Long-Range Video Reasoning"
_ICLR.cc/2026/Conference — ICLR 2026 Conference Withdrawn Submission_

### Official Review · Reviewer_R2yv · 2025-10-18

**Soundness:** 1
**Presentation:** 1
**Contribution:** 1
**Rating:** 2
**Confidence:** 3

**Summary:**

This paper introduces a training-free framework for Bengali Long-range Video Reasoning (BLrVR), evaluated on the EgoSchema benchmark. The authors further propose a new prompting strategy, termed Close-ended Answer Selection (CeAS), which incorporates structured role definitions, task-specific cues, and strict reasoning constraints to enhance LLM inference. Experimental results demonstrate that CeAS attains state-of-the-art performance, outperforming RAG methods in terms of precision, recall, and runtime efficiency, while maintaining comparable accuracy and F1-scores.

**Strengths:**

1. The paper addresses an underexplored research area by focusing on long-range video reasoning in the Bengali language, thereby contributing to the broader goal of enhancing multimodal understanding in low-resource linguistic settings.
2. The paper presents a Bengali-translated version of the EgoSchema benchmark, which, if publicly released, would serve as a valuable resource for the research community.

**Weaknesses:**

1. The paper lacks a clear technical contribution that meets the standards of ICLR. The proposed framework primarily combines existing techniques, such as CoT prompting and RAG, without introducing novel algorithmic insights or theoretical advancements relevant to the machine learning community.

2. The experimental section primarily consists of comparisons with existing methods, which makes the work appear more as an empirical evaluation than a contribution with methodological innovation.

3. Despite emphasizing the exploration of the Bengali language, the paper does not introduce any language-specific model adaptations or linguistic considerations tailored to Bengali. This weakens the central claim of advancing multimodal reasoning for low-resource languages.

4. The paper’s organization could be improved. The Method section omits critical implementation details of the proposed CeAS approach, with some necessary explanations relegated to the appendix, making it difficult for readers to fully understand and reproduce the method.

5. For qualitative examples presented in Bengali, providing English translations would enhance accessibility and clarity for a broader research audience.

**Questions:**

See weaknesses

---

### Official Review · Reviewer_UEwK · 2025-10-31

**Soundness:** 2
**Presentation:** 2
**Contribution:** 1
**Rating:** 2
**Confidence:** 3

**Summary:**

This paper presents a training-free framework for long-range video question answering in Bengali. The authors adapt the EgoSchema dataset to Bengali and propose two reasoning approaches — a prompting-based method (CeAS) and a retrieval-augmented generation (RAG) variant. The system uses visual captioning via multimodal LLMs and LLM-based reasoning to answer multiple-choice questions. Experiments compare different LLMs, captioners, and prompting schemes on the Bengali-translated EgoSchema subset.

**Strengths:**

-	Addresses a low-resource language (Bengali), which is a relevant but niche topic.
-	Long-range video reasoning is an important problem

**Weaknesses:**

-	Lack of novelty: There is no clear difference between the proposed method and other established ones like VideoAgent [1,2].
-	Limited modality: Relying only on textual captions loses visual information critical for video reasoning.
-	Outdated baselines: the baselines in experiments are quite outdated (2022, 2023).

**Questions:**

-	Can the authors clarify the difference between the proposed method with other approaches like VideoAgent [1,2]?
-	Is there any quantitative or qualitative evidence that your proposed method performs better than other recent methods?

[1] Fan, Yue, et al. "Videoagent: A memory-augmented multimodal agent for video understanding." European Conference on Computer Vision. Cham: Springer Nature Switzerland, 2024.
[2] Wang, Xiaohan, et al. "Videoagent: Long-form video understanding with large language model as agent." European Conference on Computer Vision. Cham: Springer Nature Switzerland, 2024.

---

### Official Review · Reviewer_GP3y · 2025-11-01

**Soundness:** 3
**Presentation:** 3
**Contribution:** 2
**Rating:** 4
**Confidence:** 4

**Summary:**

This paper presents BLrVR, a training-free framework for Bengali long-range video question answering.
The approach adapts the EgoSchema dataset into Bengali through machine translation and human validation, and evaluates two reasoning frameworks: 1. CeAS, a structured close-ended prompting strategy with role specification and task cues
2. RAG, a retrieval-augmented generation variant leveraging external textual evidence.

Experiments benchmark various multi-modal LLMs (Gemini-2.0/1.5, Gemma2-9B) as captioners and reasoning modules, reporting moderate gains in precision and recall for CeAS over RAG on the translated Bengali EgoSchema subset.

**Strengths:**

1. This paper adapts long-range video reasoning to Bengali is appreciated and valuable for inclusive AI research, given the paucity of multimodal datasets in South Asian languages.

2. The paper provides some empirical comparisons across captioners, LLMs, and prompting variants (CeAS, CoT, Plan-and-Solve), offering a practical reference for low-resource language setups.

3. The framework is training-free and modular, making it easy to reproduce and extend to other low-resource settings.

**Weaknesses:**

1. Limited technical novelty: I think the core contributions (CeAS prompt design, RAG baseline) are largely incremental and primarily involve reusing existing prompting and retrieval frameworks in a Bengali setting. The novelty lies more in application and data translation rather than in algorithmic innovation.

2. Dataset contribution is minimal: The “Bengali EgoSchema” is simply a translated version of an existing benchmark with limited linguistic validation; there is no evidence of new video content, annotation schema, or task definition.

also, results hover around 69% accuracy, with small differences between methods. The evaluation lacks ablations on reasoning depth, temporal grounding fidelity, or cross-language generalization, which would be necessary for an ICLR-style contribution + missing important comparison/discussion with the line of related work [1-4] (and more).


Overall, I feel the focus on prompt structure, translation quality, and linguistic adaptation aligns more naturally with ACL/EMNLP tracks (e.g., multilingual or low-resource multimodal reasoning), rather than ICLR’s emphasis on technical advances in learning representations or modeling architectures.


[1] A simple llm framework for long-range video question-answering.
[2] Videotree: Adaptive tree-based video representation for llm reasoning on long videos.
[3] Lifelongmemory: Leveraging llms for answering queries in long-form egocentric videos.
[4] VideoLucy: Deep Memory Backtracking for Long Video Understanding

**Questions:**

Please see the weakness section, overall, I have doubts about the area of contribution is aligned with ICLR requirement or not.

---

### Note · Authors · 2025-12-25

I have read and agree with the venue's withdrawal policy on behalf of myself and my co-authors.